# The increase in cell volume and nuclear number of the koji-fungus *Aspergillus oryzae* contributes to its high enzyme productivity

Ayaka Itani[1], Haruto Motomura[1], Ken Oda[2], Hideyuki Yamashita[3], Kanae Sakai[4], Ken-ichi Kusumoto[4], Shinsuke Shigeto[5], Takehiko Ichikawa[6], Hosain Mohammad Mubarak[6], Takeshi Fukuma[6], Takuya Katayama[7,8], Jun-ichi Maruyama[7,8], Shunsuke Masuo[1], Naoki Takaya[1], Norio Takeshita[1]*

[1]Microbiology Research Center for Sustainability (MiCS), Faculty of Life and Environmental Sciences, Tsukuba Institute for Advanced Research (TIAR), University of Tsukuba, Tsukuba, Japan; [2]National Research Institute of Brewing, Higashi-hiroshima, Japan; [3]Higuchi Matsunosuke Shoten Co., Ltd., Osaka, Japan; [4]Department of Biotechnology, Graduate School of Engineering, Osaka University, Osaka, Japan; [5]Department of Chemistry, School of Science, Kwansei Gakuin University, Hyogo, Japan; [6]Nano Life Science Institute (WPI-NanoLSI), Kanazawa University, Kanazawa, Japan; [7]Department of Biotechnology, The University of Tokyo, Tokyo, Japan; [8]Collaborative Research Institute for Innovative Microbiology, The University of Tokyo, Tokyo, Japan

*For correspondence: takeshita.norio.gf@u.tsukuba. ac.jp

## eLife Assessment

The ratio of nuclei to cell volume is a well-controlled parameter in eukaryotic cells. This study now reports **important** findings that expand our understanding of the regulatory relationship between cell size and number of nuclei. The evidence supporting the conclusions is **convincing** obtained by applying appropriate and validated methodology in line with current state-of-the-art. The paper will be of broad interest for cell biologists and fungal biotechnologists seeking to understand mechanisms determining cell size and number of nuclei and why this knowledge might also be of importance for the production of enzymes and thus production strains not only of Aspergillus oryzae but also other industrially used fungi.

**Abstract** While the ratio of nuclei to cell volume is well regulated, it remains largely unexplored in multinucleate organisms. The koji-fungus *Aspergillus oryzae*, traditionally used in Japanese brewing and fermentation for over a thousand years, is now widely utilized in modern biotechnology as a host for enzyme production. We discovered that, over time in culture, hyphae become thicker, resulting in a tenfold increase in cell volume, and the number of nuclei in hyphal cells also increases tenfold, exceeding 200. The increase in cell volume and nuclear number is unique among the investigated *Aspergillus* species and correlates with its high enzyme production capabilities. Since nuclear number and cell volume are correlated, both must increase simultaneously for either to expand. Our analyses identified genetic factors and nutritional environmental signals involved in each of these increases. Increases in nuclear number and cell volume were also observed in other fungi bred for industrial use. This study not only deepens our understanding of the evolutionary processes that promote high enzyme productivity through fungal breeding, but also provides

insights into the molecular mechanisms regulating cell volume and nuclear number in multinucleate organisms.

## Introduction

In unicellular organisms, cell volume is tightly regulated in coordination with nuclear division timing (*Amodeo and Skotheim, 2016*; *Nurse, 1975*; *Jorgensen et al., 2007*). Similarly, in multinucleated filamentous fungi, a correlation exists between cell volume and nuclear number (*Dynesen and Nielsen, 2003*; *Dundon et al., 2016*). However, the number of nuclei varies across species (*Roper et al., 2011*), the factors governing nuclear number and cell volume remain largely unknown. Filamentous fungi are industrially significant microorganisms, and the strains selected through breeding provide ideal research models for studying these evolutionary processes.

Filamentous fungi secrete a variety of hydrolytic enzymes to support their growth, and humans have long harnessed this enzyme secretion ability for the production of a wide range of fermented foods and beverages (*Hurst, 2023*). *Aspergillus oryzae* has been used in Japan for over a thousand years to produce traditional fermented foods like sake, soy sauce, and miso (*Kitamoto, 2015*; *Ichishima, 2016*). A distinctive feature of the fermentation techniques is solid-state cultivation using substrates like rice, soybeans, and wheat bran. The inoculum of *A. oryzae* used in fermentation, known as 'koji', has been commercially produced for around 700 years (*Yamashita, 2021*). Koji is created by cultivating koji-fungus on grains, and koji starter manufacturers have selected and bred strains to improve the flavor and color of various fermented products. The *A. oryzae* strains currently in use are mainly preserved and managed by koji manufacturers and research institutions. A notable characteristic of *A. oryzae* is its high ability to secrete starch-degrading enzymes and proteases, which has led to extensive research into its genome and gene expression regulation (*Machida et al., 2005*; *Machida et al., 2008*; *Gomi, 2019*; *Tanaka and Gomi, 2021*; *Wang et al., 2010*; *Ishida et al., 2000*; *Kitano et al., 2002*).

Although *A. oryzae* shares 99.5% genomic similarity with *Aspergillus flavus*, which produces the carcinogenic aflatoxin (*Rokas et al., 2007*), there have been no reports of *A. oryzae* producing aflatoxins, and its safety has been confirmed at the molecular level (*Kusumoto et al., 2000*; *Kobayashi et al., 2007*). Phylogenomic analyses suggest that *A. oryzae* and *A. flavus* diverged approximately 50,000–189,000 years ago (*Watarai et al., 2019*; *Chang and Ehrlich, 2010*). Over the following millennia, human use of *A. oryzae* is thought to have driven significant genomic recombination, resulting in its evolution into a 'cell factory' specifically optimized for the breakdown of sugars and proteins (*Gibbons et al., 2012*).

Additionally, modern biotechnology utilizes filamentous fungi as cell factories to produce organic acids, enzymes, and pharmaceuticals (*Meyer et al., 2016*; *Liu et al., 2023*; *Meyer et al., 2020*). *A. oryzae* also shows high production capacity as a host for both homologous and heterologous protein production in modern biotechnology (*Christensen et al., 1988*; *Saito et al., 2024*). Recently, its application has expanded to include the production of pharmaceutical proteins and secondary metabolites (*Huynh et al., 2020*; *Itoh et al., 2010*). Furthermore, filamentous fungi involved in fermentation play a crucial role in creating a more sustainable food system, such as by upcycling agricultural by-products into food through fungal fermentation and using *A. oryzae* mycelium for alternative meats (*Maini Rekdal et al., 2024b*; *Maini Rekdal et al., 2024a*).

While basic and applied research on *A. oryzae* continues, fundamental questions remain unanswered, such as why *A. oryzae* exhibits high enzyme production capacity and why it excels as a host for heterologous expression. This study shows that the key lies in the increase in cell volume and nuclear number and analyzes the molecular mechanisms.

## Results

### Increase in nuclear number and cell volume in *A. oryzae*

In the model strain *A. oryzae* RIB40, hyphae on day 1 of cultivation contain 10–20 nuclei per cell, but by day 2 of cultivation, apical cells with more than 200 nuclei appear (*Yasui et al., 2020*). We classified the hyphae into three categories based on the number and distribution of nuclei and compared their proportions after 1–3 days of cultivation (see methods, images were shown previously) (*Yasui et al.,*

*2020*). We found that the proportion of hyphae with a higher number of nuclei (represented by class III, over 200 nuclei in the cell) increased as the culture period increased from 1 to 3 days (*Figure 1A, B*; *Yasui et al., 2020*). This phenotype was not observed in the model fungus *Aspergillus nidulans* or the closely related species *A. flavus* (*Figure 1A, B*; *Yasui et al., 2020*). As far as we know, such phenotypes have not been observed in other strains of *A. nidulans*, *Aspergillus niger*, *Aspergillus fumigatus*, or other strains commonly used in research. Among other *Aspergillus* species used in fermentation (*Ichishima, 2016*; *Yamashita, 2021*), the proportion of hyphae with a higher number of nuclei increased in *Aspergillus sojae*, which is used for miso and soy sauce production. This phenotype was not observed in *Aspergillus luchuensis* nor in its albino mutant *Aspergillus luchuensis mut. kawachii*, which are used for distilled spirits such as awamori and shochu. Among several strains of *A. oryzae* used in industrial applications, the RIB915 strain used for soy sauce fermentation does not show an increase in the number of nuclei, while the RIB128 and RIB430 strains used in sake brewing exhibit a significant increase in the number of nuclei, suggesting that different phenotypes were selected through distinct breeding processes.

The distribution of nuclei in the hyphae of *A. oryzae* RIB40 was visualized through 3D reconstruction (*Figure 1C*, *Video 1*). In the class I hyphae, the nuclei were spaced relatively evenly, whereas in the class III hyphae with increased nuclei, the nuclei were densely packed and arranged in a disordered manner. Furthermore, the nuclear size in the class III hyphae was significantly smaller than that in the class I hyphae (*Figure 1D, E*). Nuclear division is synchronized in the hypha even when there are more than 200 nuclei (*Yasui et al., 2020*), suggesting that DNA replication occurs similarly in most nuclei. The germination rate of conidia and the colonies formed from individual conidia show no significant abnormalities, suggesting that nearly all nuclei possess normal genomes and chromosomes.

Strains without an increase in nuclear number had hyphae with diameters ranging from 2 to 7 μm, while strains with an increase in nuclear number had some hyphae of similar thickness but often exhibited thicker hyphae exceeding 10 μm (*Figure 1F*). We compared the number of nuclei and cell volume within the first 100 μm from the hyphal tip, due to variation in the distance from the hyphal tip to the septum, among *A. oryzae* strains (RIB40, RIB128, and RIB915) and *A. flavus* (*Figure 1G*). A strong positive correlation was observed between number of nuclei and cell volume. This is consistent with previously reported that the number of nuclei per hyphal volume remains constant (*Dynesen and Nielsen, 2003*; *Dundon et al., 2016*).

## Thick hyphae with increased nuclei emerge by branching

Time-lapse imaging using the *A. oryzae* RIB40 expressing H2B-GFP indicated that thick hyphae with increased nuclei emerged by branching from hyphae without increased nuclei (*Figure 2A*, *Video 2*). In the emerged thick hyphae, rapid nuclear division occurred at the sites of branching (*Figure 2B*, *Video 3*). Immediately after branching, the thick hyphae showed a significantly faster rate of nuclear proliferation, increasing four to five times within 8 hr compared to the thin hyphae immediately after branching (*Figure 2C*). Branches from hyphae with increased nuclei produced both hyphae with and without increased nuclei (*Figure 2D*).

Time-lapse imaging was conducted over a wider area, and image processing was applied to color-code the nuclei: green for those in hyphae with increased nuclei and white for those in hyphae without increased nuclei (*Figure 2E*, *Figure 2—figure supplement 1A*, *Video 4*, see methods). Up to 24 hr after inoculation, most hyphae were thin with no increase in nuclei number, but thick hyphae with increased nuclei began to appear as time progressed (*Figure 2E, F*). Since thick hyphae elongated faster than thin hyphae (*Figure 2G*), thick hyphae outgrew and overtook thin hyphae, eventually dominating the colony perimeter (*Figure 2E, F*).

Transmission electron microscopy confirmed basic nuclear structures, such as the nuclear membrane and nucleolus, in both thick and thin hyphae (*Figure 2—figure supplement 1B*). As hyphal diameter increased, cell walls became more uneven and thicker (*Figure 2H, I*). This finding corresponds with Raman spectroscopy results showing pronounced peaks corresponding to cell wall polysaccharides in thick hyphae (*Figure 2—figure supplement 1C*; *Hossain et al., 2023*). Additionally, peaks associated with active mitochondria were prominent in thick hyphae (*Figure 2—figure supplement 1C*; *Yasuda et al., 2019*), which was further confirmed by fluorescence staining (*Figure 2—figure supplement 1D*). The surface structure and mechanical properties of the cell wall were analyzed using atomic force microscopy (AFM). Thick hyphae exhibited greater surface

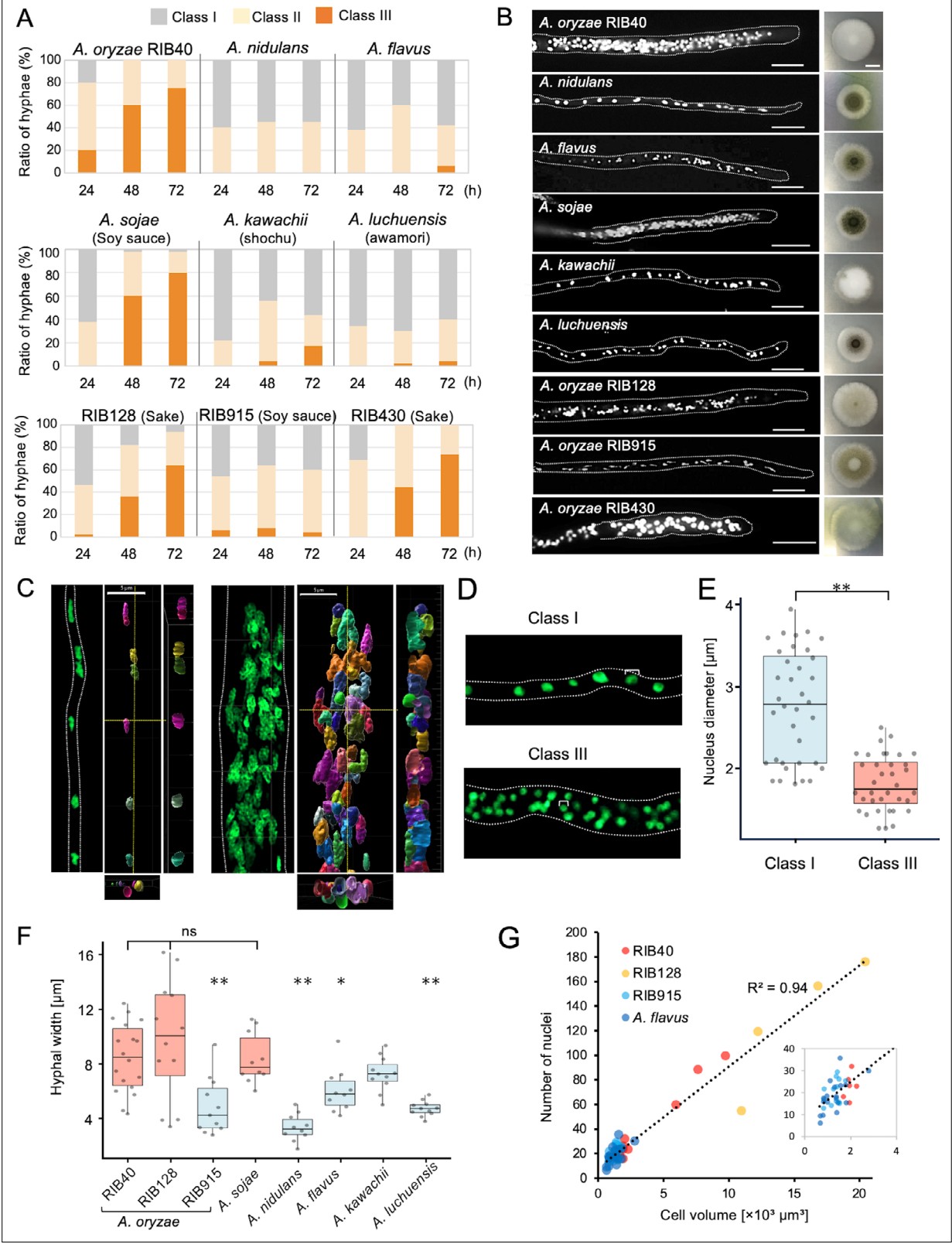

**Figure 1.** Increase in number of nuclei and cell volume in *Aspergillus* species. (**A**) The nuclear distribution in the tip cells was categorized into classes I–III. The ratio of hyphae in each class was measured in *Aspergillus* species at 24, 48, and 72 hr of growth (*n* = 50). Data for *A. oryzae* RIB40 and *A. nidulans* are reproduced from a previous study (*Yasui et al., 2020*). (**B**) Colony morphology after 3 days of culture on the minimal medium. The nuclear distribution in the hyphae at the colony periphery stained with SYBR Green. Scale bars: hyphae, 20 μm; colonies, 1 cm. (**C**) 3D images of hyphae without

*Figure 1 continued on next page*

*Figure 1 continued*

increased nuclei (left) and with increased nuclei (right) in *A. oryzae* RIB40 from *Video 1*. Each nucleus is indicated with different colors by Imaris soft. Scale bar: 5 μm. (**D**) Comparison of nuclear size in class I hyphae (2.4 μm) and class III hyphae (1.6 μm), indicated by white lines. (**E**) Box plots of nucleus diameters in hyphae without increased nuclei and those with increased nuclei (*n* = 35, **p < 0.01, *t*-test). (**F**) Box plots of hyphal width at the colony periphery grown for 72 hr (*n* = 10–14, **p < 0.01, *p < 0.05, *t*-test). Strains with increased nuclei are marked in red, and those without are marked in blue. (**G**) Correlation between nuclear number and cell volume of hyphae at 100 μm from the tips in *A. oryzae* RIB40, RIB128, RIB915, and *A. flavus*.

roughness and higher elasticity compared to thin hyphae (*Figure 2J, K*, *Figure 2—figure supplement 1E, F*).

## Increase in nuclear number and enzyme secretion

We compared the amounts of secreted proteins in *A. oryzae* RIB40, RIB128, RIB915, *A. nidulans*, and *A. flavus* (*Figure 3A*). The strains that show an increase in nuclear number had significantly higher levels of protein secretion than the strains that do not. In this condition, α-amylases are the dominant proteins secreted by *A. oryzae* (*Wang et al., 2010*). The amylase enzyme activity was compared between *A. oryzae* RIB40, which shows an increase in nuclear number, and *A. oryzae* RIB915, which does not show an increase. In RIB40, starting from day 2, when hyphae with increased nuclear numbers appeared, the enzyme activity per fungal biomass significantly increased more than RIB915 (*Figure 3B*).

To compare the amylase activity between hyphae with a higher and a lower number of nuclei, amylase activity was monitored under conditions where a single hypha grew in a microfluidic channel with a fluorescent substrate (*Figure 3C*, *Figure 3—figure supplement 1A*, *Video 5*; *Itani et al., 2023*). After 12 hr, the fluorescence intensity in the channel with thick hyphae was about three times higher than that in the channel with thin hyphae, indicating that thicker hyphae secreted more amylase than thinner hyphae (*Figure 3D*).

We found that the addition of yeast extract increased both hyphal width and nuclear number in the RIB915 strain (*Figure 3B*, *Figure 3—figure supplement 1E*). In *A. oryzae* RIB40, the addition of yeast extract did not make the already thick hyphae any thicker but increased the proportion of thicker hyphae. The addition of yeast extract increased protein secretion approximately 1.8 times in RIB40 and 8.5 times in RIB915 (*Figure 3F*). Although the addition of yeast extract did not cause a dramatic increase in nuclear number in *A. nidulans*, hyphal width increased by 1.4 times and protein secretion increased by 5.1 times.

The addition of nucleic acids, peptone, or casamino acids to the minimal medium increased the number of nuclei in *A. oryzae* RIB915, while vitamin B did not cause a significant change in nuclear number (*Figure 3—figure supplement 1C*). Although the addition of individual amino acids did not show the same effect as yeast extract, the addition of asparagine, proline, glutamine, and branched-chain amino acids increased the proportion of hyphae with increased nuclei by 40–50% (*Figure 3G*, *Figure 3—figure supplement 1D*). The amino acids that induced nuclear number increase also led to an increase in protein secretion, indicating a positive correlation between the ratio of class III hyphae and the increase in secreted proteins (*Figure 3H*, *Figure 3—figure supplement 1E*). Since the amino acids inducing an increase in nuclear number may activate the Target of Rapamycin (TOR) pathway (*Wouters and Koritzinsky, 2008*), *A. oryzae* RIB40 was grown with low concentration of rapamycin, a TOR pathway inhibitor, which minimally affects colony size. Under the conditions, RIB40 did not produce hyphae with an increased number of nuclei (*Figure 3I–K*, *Figure 3—figure supplement 1F*). Rapamycin decreased the ratio of hyphae with increased nuclei even in the medium with yeast extract (*Figure 3—figure supplement 1G*).

## Transcriptome analyses in hyphae with increased nuclei

To investigate gene expression changes in hyphae with increased nuclei, transcriptome analysis was

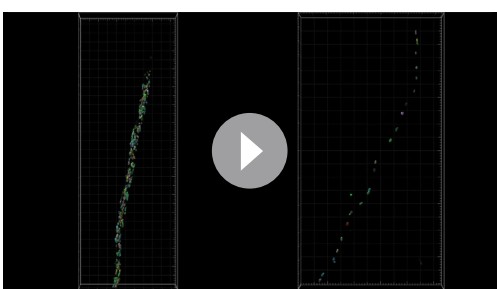

**Video 1.** 3D images of hyphae without increased nuclei (left) and with increased nuclei (right) in *A. oryzae* RIB40. Each nucleus is indicated with different colors by Imaris soft. Scale bar: 5 μm.

https://elifesciences.org/articles/107043/figures#video1

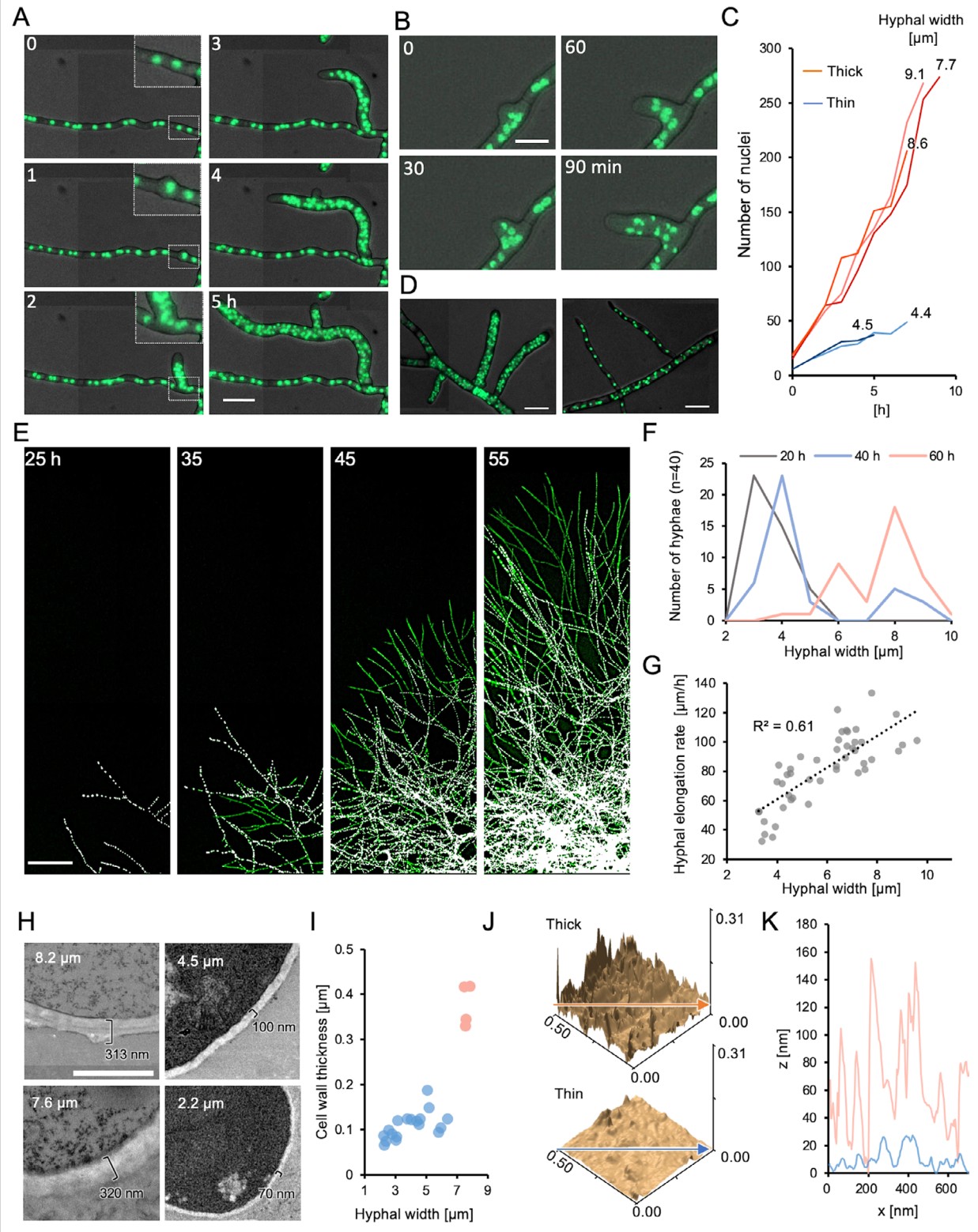

**Figure 2.** Thick hyphae with increased nuclei emerge by branching. (**A**) Time-lapse image sequence of *A. oryzae* RIB40 expressing H2B-GFP showing the emergence of thick hyphae with increased nuclei by branching from *Video 2*. Scale bar: 20 µm. Elapsed time is indicated in minutes. (**B**) Image sequence of successive nuclear division within the newly emerged branched hypha from *Video 3*. Scale bar: 10 µm. Elapsed time is indicated in minutes. (**C**) Time course of nuclear number per hypha. Nuclear numbers were counted with the start time of branching set as 0, every hour for 5–10 hr. Thick: hyphal width >7 µm, Thin: hyphal width <5 µm. (**D**) Branching from hyphae with increased nuclei generates both thick hyphae with increased

*Figure 2 continued on next page*

*Figure 2 continued*

nuclei (left) and thin hyphae without increased nuclei (right). Scale bars: 20 µm. (**E**) Image sequence of mycelial growth showing hyphae with increased nuclei (green) and without increased nuclei (white) from *Video 4*. Scale bars: 200 µm. Elapsed time is indicated in hours. (**F**) Line histogram of hyphal width at the colony periphery at 20, 40, and 60 hr, calculated from *Video 4* (n = 40). (**G**) Correlation between hyphal width and maximum elongation rate calculated from *Video 4* (n = 45). The maximum elongation rate is determined from elongation rates measured every hour over a 10-hr period. (**H**) Transmission electron microscopy (TEM) images of *A. oryzae* RIB40 hyphae. Hyphal diameter and cell wall thickness are indicated in white and black text, respectively. Scale bar: 1 µm. (**I**) Correlation plots of hyphal width and cell wall thickness based on TEM images. Thick hyphae (>7 µm) are marked in red, and thin hyphae (<7 µm) are marked in blue. (**J**) 3D surface images of *A. oryzae* RIB40 hyphal tips in the thick hypha (upper) and the thin hypha (lower) constructed using atomic force microscopy (AFM). (**K**) Surface roughness of cell walls in thick (red) and thin (blue) hyphae along the arrows in J.

The online version of this article includes the following figure supplement(s) for figure 2:

**Figure supplement 1.** Comparative characterization of thick and thin hyphae.

---

performed under six conditions: In minimal medium, *A. oryzae* RIB40, which shows increased nuclei, *A. oryzae* RIB915 and *A. nidulans*, which do not show increased nuclei. In minimal medium with yeast extract, *A. oryzae* RIB40 and RIB915, which show increased nuclei, and *A. nidulans*, which does not show increased nuclei (*Figure 4A*, *Figure 3—figure supplement 1B*). In RIB915, supplementation with yeast extract led to a more than fourfold increase in the expression of 660 genes, among which 449 genes did not show such increase in RIB40 or *A. nidulans* (*Figure 4B*, *Supplementary file 1*). Gene ontology (GO) analysis of these genes revealed that processes related to cell wall synthesis and divalent metal ion transport were significantly enriched (*Figure 4C*).

To capture gene expression changes associated with increased nuclei with higher resolution, thick and thin hyphae from RIB40 were dissected using laser microdissection, collected separately, and subjected to transcriptome analysis (*Figure 4D*, *Video 6*). Clear differences in gene expression pattern were observed between thick and thin hyphae (*Figure 4E*). 558 genes that were more than fourfold upregulated in thick hyphae but not in thin hyphae (*Supplementary file 2*). GO analysis of the upregulated genes indicated an enrichment of processes related to biogenesis of ribosome and cellular component (*Figure 4F*). Among them, 21 matched those from (*Figure 4G*, *Supplementary file 3*). Of these, 13 had GO annotations, which were related to conidiation, cell wall synthesis, Ca$^{2+}$ transport, and rRNA processing (*Figure 3H*). KEGG map visualization indicates that overall ribosomal gene expression increased in the thick hyphae than the thin hyphae (*Figure 4—figure supplement 1A*).

Among the gene list, we focused on *msy1* and *msy2*, which are involved in Ca$^{2+}$ transport and maintaining cell volume homeostasis in *Schizosaccharomyces pombe* (*Nakayama et al., 2012*; *Nakayama et al., 2014*). Disruption of the ortholog genes, *msyA* and *msyB*, in *A. oryzae* RIB40 did not have a significant impact on colony growth, hyphal morphology, and number of nuclei in minimal medium (*Figure 4—figure supplement 1B*). Under hypoosmotic shock conditions, the △*msyA* often indicated cell lysis near the hyphal tips (*Figure 4I, J*). Under hypoosmotic conditions, water influx into the cells causes hyphal cell expansion, but the volume and turgor pressure are regulated to prevent cell lysis. The *msyA* disruption, however, impairs the regulation of the hyphal cell expansion, leading to hyphal lysis (*Figure 4—figure supplement 1C–E*).

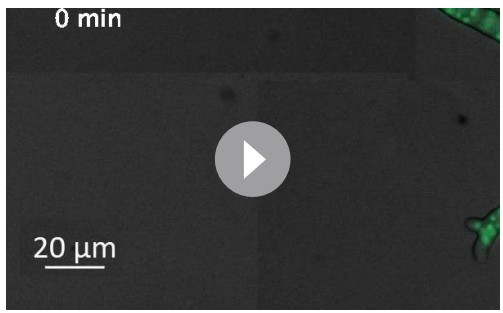

**Video 2.** Emergence of thick hyphae with increased nuclei by branching in *A. oryzae* RIB40 expressing H2B-GFP. Scale bar: 20 µm. Elapsed time is indicated in minutes.

https://elifesciences.org/articles/107043/figures#video2

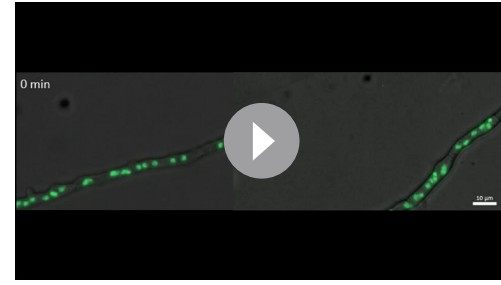

**Video 3.** Successive nuclear division within the newly emerged thick branched hypha in *A. oryzae* RIB40 expressing H2B-GFP. Scale bar: 10 µm. Elapsed time is indicated in minutes.

https://elifesciences.org/articles/107043/figures#video3

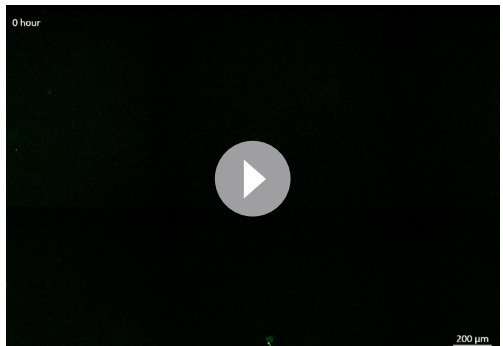

**Video 4.** Mycelial growth of hyphae with increased nuclei (green) and without increased nuclei (white). Scale bars: 200 µm. Elapsed time is indicated in hours. https://elifesciences.org/articles/107043/figures#video4

## SNP analysis among industrial strains with distinct phenotypes

The fact that currently used *A. oryzae* strains are managed and preserved by koji manufacturers and research institutions is a notable advantage of *A. oryzae* research. Whole-genome sequencing and comparative genome analysis of 82 *A. oryzae* strains with different applications have shown that these strains cluster into several clades (*Meyer et al., 2016*). We investigated whether nuclear numbers increase in 20 strains selected from six different clades (*Figure 5A*). In the minimal media supplemented with yeast extract, all strains exhibited an increase in nuclear number (*Figure 5— figure supplement 1A*). In the minimal media without yeast extract, three strains from Clade C showed an increase in nuclear number, whereas strains from Clades A, B, and E did not (*Figure 5A, B*, *Figure 5—figure supplement 1A*). Clades F and G included both strains with increased nuclear numbers and strains without. SNP analysis of ORFs in clades F and G revealed 108 mutations in clade F between TK-32 and TK-38, and 23 mutations in clade G between TK-41 and TK-47 (*Supplementary file 4 and 5*).

Among these, AO090038000626; *rseA*, predicted glycosyl transferase, was the only common gene. This gene has been identified as a mutation in an *A. sojae* mutant with high extracellular enzyme production (*Ogawa et al., 2021*). It is also shown that deletion of *rseA* in *A. nidulans* leads to cell wall defects and promotes enzyme secretion (*Ogawa et al., 2021*). Comparison of mutations in *rseA* among strains in clades C, F, and G indicates a few amino acid substitutions within the ORF, but no common mutations corresponded to the phenotype (*Figure 5C*, *Figure 5—figure supplement 1B*). Structural prediction suggests that the mutations are located inside the enzyme domain (*Figure 5— figure supplement 1C*).

The disruption of the *rseA* in *A. oryzae* RIB40 resulted in significant growth delay, increased branching, and the absence of hyphae with increased nuclei (*Figure 5D*, *Figure 5—figure supplement 1D*). The *rseA* gene DNA fragments from the strains with increased nuclei (RIB40, TK-47) or from the strain without increased nuclei (TK-41) were amplified by PCR and introduced ectopically into RIB915 (*Figure 5E*). The strains expressing *rseA* from RIB40 or TK-47 exhibited an increase in hyphal width and a corresponding increase in the proportion of hyphae with more nuclei, whereas the strain expressing *rseA* from TK41 showed no increase in hyphal width or nuclear number (*Figure 5E, F*, *Figure 5—figure supplement 1E*).

## Industrial breeding strains from other genera

To investigate whether the phenomenon of increased nuclear numbers also applies to other industrial fungi, we compared the bred strain of *Trichoderma reesei* (QM9414) for cellulase production and its control strain (QM6a) (*Vitikainen et al., 2010*), as well as the bred strain of *Penicillium chrysogenum* (IFO4688) for penicillin production and wild-type strain (*Barreiro et al., 2012*). In minimal medium, no significant differences in hyphal width or nuclear number were observed between the bred and control strains, whereas with the addition of yeast extract, a notable increase in nuclear number and hyphal width was observed only in the bred strains (*Figure 5G, H*, *Figure 5—figure supplement 2A–C*). The addition of yeast extract significantly increased cellulase and protease activity only in the bred strain of *T. reesei* and *P. chrysogenum*, respectively (*Figure 5I, J*). These results suggest that even in modern biotechnology, where high enzyme or compound producing strains are bred, strains with increased nuclear numbers were selected, like the traditional breeding practices of koji-fungi.

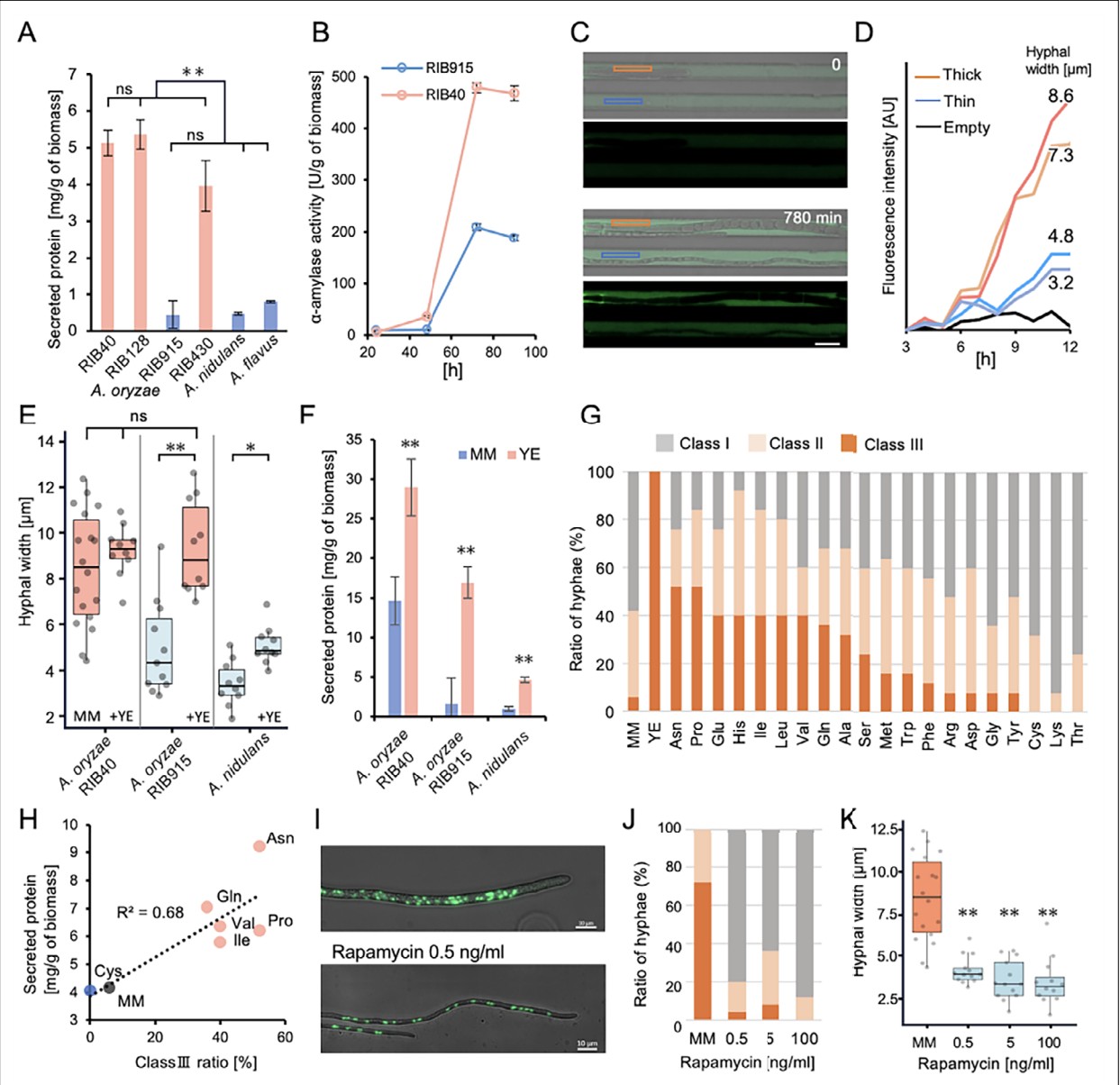

**Figure 3.** Correlation between number of nuclei and enzyme secretion. (**A**) Secreted protein per biomass in strains with increased nuclei (red) and without increased nuclei (blue) after 4 days of culture in minimal medium with 1% maltose (mean ± SE, $n = 3$, **$p < 0.01$, $t$-test). (**B**) Time course of α-amylase activity per biomass in *A. oryzae* RIB40 and RIB915 cultured in minimal medium with 1% maltose (mean ± SE, $n = 3$). (**C**) Measurement of α-amylase activity in a single hypha using a microfluidic device. A fluorescent substrate increases in fluorescence upon hydrolysis by α-amylase. ROIs for thick hypha are marked in red, and thin hypha in blue. Scale bar: 10 μm. (**D**) Temporal changes in fluorescence intensity measured in individual flow channels, as described in C. (**E**) Box plots of hyphal width in colonies grown in minimal medium with or without 1% yeast extract ($n = 10$–18, **$p < 0.01$, *$p < 0.05$, $t$-test). Conditions with increased nuclei are marked in red, and those without in blue. (**F**) Secreted protein per biomass in minimal medium with or without 1% yeast extract (mean ± SE, $n = 3$, **$p < 0.01$, $t$-test). (**G**) Ratio of class I–III hyphae in *A. oryzae* RIB915 colonies grown in minimal medium supplemented with 1% yeast extract or 0.1% individual amino acids ($n = 50$). (**H**) Correlation between the ratio of Class III hyphae and secreted protein per biomass in the 0.1% amino acid-supplemented medium. (**I**) Images of hyphae and nuclear distribution in *A. oryzae* RIB40 grown on minimal medium with or without 0.5 ng/ml rapamycin. Scale bars: 10 μm. (**J**) Ratio of class I–III hyphae in *A. oryzae* RIB40 cultured on minimal medium containing 0, 0.5, 5, or 100 ng/ml rapamycin ($n = 50$). (**K**) Box plots of hyphal width in the colonies cultured under the conditions in J ($n = 11$–18, **$p < 0.01$, $t$-test).

The online version of this article includes the following figure supplement(s) for figure 3:

**Figure supplement 1.** Single-hypha enzyme activity assay and the influence of YE, amino acids, and rapamycin on nuclear increase.

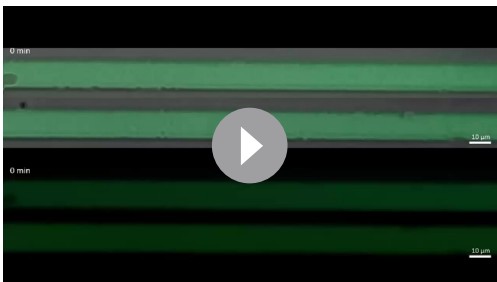

**Video 5.** Amylase activity was monitored with a fluorescent substrate under conditions where a single thick or thin hypha grew in a microfluidic channel. Scale bar: 10 μm. Elapsed time is indicated in minutes. https://elifesciences.org/articles/107043/figures#video5

## Discussion

In *A. oryzae* some strains, hyphae grow about three times thicker and contain roughly ten times more nuclei than the initial hyphae (*Figure 1*). These thicker hyphae emerge from existing branches, where nuclear division is promoted, resulting in the formation of hyphae with an increased number of nuclei (*Figure 2*). A clear correlation was observed between hyphal cell volume and nuclear number (*Figure 1G*). This consistency aligns with findings in other organisms, from unicellular yeast to multicellular plants and animals, where the nuclear-to-cytoplasmic ratio is tightly regulated, allowing cell cycle progression only upon reaching a critical cell size (*Amodeo and Skotheim, 2016*; *Nurse, 1975*; *Jorgensen et al., 2007*). Both the G1/S and G2/M transitions are controlled by size checkpoints, with specific mechanisms differing between species (*Futcher and Kellogg, 2024*; *Fantes and Nurse, 1977*).

In *A. nidulans*, it is known that the duration of the G1 and G2 phases is regulated by temperature (*Bergen and Morris, 1983*). While the mechanisms governing the G1/S phase transition remain unclear (*Dörter and Momany, 2016*), the G2/M phase transition has been shown to involve the following: During the interphase of *A. nidulans*, NimX (CDK1) binds to NimE (cyclin B), and its activity is regulated through phosphorylation by AnkA (Wee1 kinase) and NimT (Cdc25 phosphatase) (*Gould and Nurse, 1989*; *Wu et al., 1998*; *Ye et al., 1997*). The activated NimX–NimE complex subsequently dephosphorylates by NimT and activates NimA (Never-in-Mitosis A) protein kinase (*Shen and Osmani, 2013*; *Osmani et al., 1987*). NimA is essential for the transition from G2 to mitosis, and its expression is tightly regulated during the cell cycle (*Osmani et al., 1987*). The mRNA level of *nimA* and *nimT* begins to increase in G2, reaches a plateau during late G2 and M phases, and is sharply degraded at the end of mitosis upon re-entry into interphase (*Osmani et al., 1987*; *Osmani et al., 1991*), meaning high expression levels of *nimA* and *nimT* result in a longer G2 phase. In thick hyphae, *nimA* and *nimT* mRNA was hardly detected, suggesting that thick hyphae sense cell volume in some way and subsequently shorten the G2 phase (*Figure 6A*). The expression level of *nimE* was low in both thick and thin hyphae, with no significant difference observed. As known in other organisms, its function is likely regulated through phosphorylation and protein degradation.

Branching is thought to occur through the degradation and reconstruction of the cell wall at the branching site (*Harris, 2008*). SNP analysis between industrial strains with different phenotypes identified RseA, a predicted glycosyltransferase (*Figure 5*). Deletion of *rseA* caused defects in cell wall synthesis in *A. nidulans*, leading to increased enzyme secretion (*Ogawa et al., 2021*; *Feng et al., 2017*). Although the targets of RseA are unknown, RseA could be involved in the cell wall integrity by modifying glycosyl chains on enzymes responsible for cell wall synthesis and degradation. Mutations in *rseA* may impair the regulation of cell wall loosening at the branching site, resulting in the formation of thicker hyphae. In fact, in the thick hyphae of RIB40, localized imbalances in cell wall components have been demonstrated by transmission electron microscopy and AFM.

Thicker hyphae can only form if high turgor pressure is maintained within the hyphae, in addition to changes in cell wall remodeling during branching. *msyA* and *msyB*, whose orthologues are calcium channels involved in the regulation of cell volume, were identified among the genes that show a significant increase in expression in thicker hyphae (*Figure 4*; *Nakayama et al., 2012*; *Nakayama et al., 2014*). Analysis of the knockout strains demonstrated the function of cell volume regulation adapted to increased turgor pressure in hypoosmotic conditions. The function of these calcium channels is likely crucial for maintaining the balance of turgor pressure and cell volume.

The TOR pathway is a central regulator of cellular growth, responding to a variety of nutrients, such as amino acids, and carbon sources, as well as environmental factors like oxygen levels, osmotic pressure, pH, and temperature stress (*Urban et al., 2007*; *De Virgilio and Loewith, 2006*). Upon sensing these inputs, the TOR pathway influences translation activity, metabolic flow, and cell size (*De Virgilio*

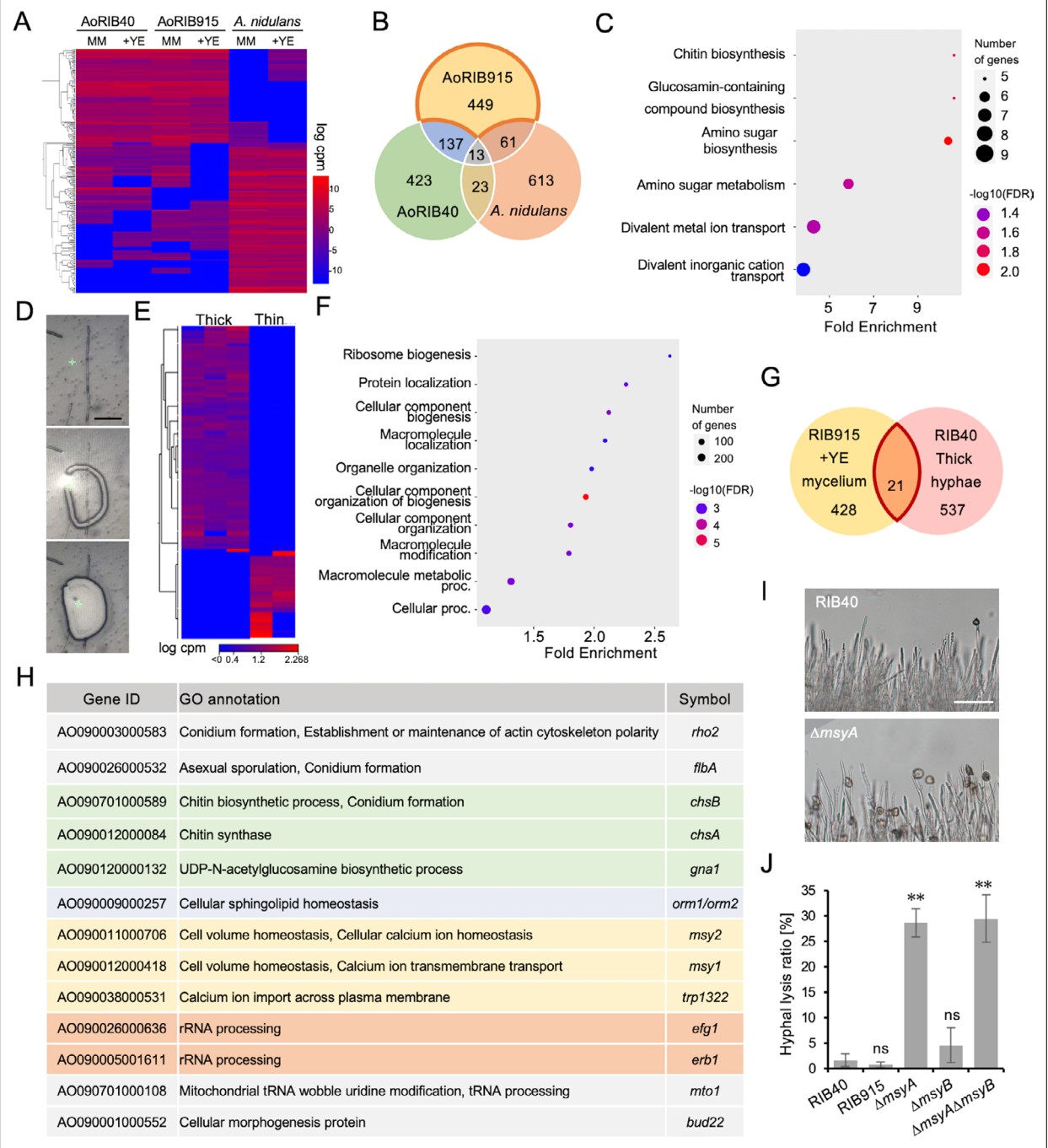

**Figure 4.** Transcriptome analyses in hyphae with increased nuclei. (**A**) Heatmap of gene expression in *A. oryzae* RIB40, RIB915, and *A. nidulans* grown in the minimal medium with or without 1% yeast extract. The top 500 genes with the largest variation in expression are shown. (**B**) Venn diagram of genes upregulated more than fourfold in the medium with yeast extract. (**C**) Gene ontology (GO) process analysis of genes uniquely upregulated in *A. oryzae* RIB915 grown in the medium with yeast extract. (**D**) Image sequence of cutting and collecting the targeted hypha by using laser microdissection. Scale bar: 100 μm. (**E**) Heatmap of gene expression in thick (*n* = 3) and thin (*n* = 2) hyphae dissected from *A. oryzae* RIB40. The top 300 genes with the largest expression differences are shown. (**F**) GO process analysis of genes upregulated more than fourfold in thick hyphae compared to thin hyphae. (**G**) Venn diagram of 449 genes upregulated in RIB915 with yeast extract and 558 genes upregulated in thick hyphae of RIB40. (**H**) GO annotations of 13 genes from the 21 shared genes in G. Genes related to cell wall processes are shown in green, cell membrane in blue, Ca²⁺ transport in yellow, and rRNA processing in orange. (**I**) Images of hyphae of RIB40 and Δ*msyA* strains after 3 min of low osmotic stress. Scale bar: 200 μm. (**J**) Ratio of hyphal tip lysis under the hypoosmotic shock (mean ± SE, *n* = 50, \*\*p < 0.01, *t*-test).

The online version of this article includes the following figure supplement(s) for figure 4:

**Figure supplement 1.** Expression changes of ribosomal genes and phenotypes of msyA and msyB deletion mutants.

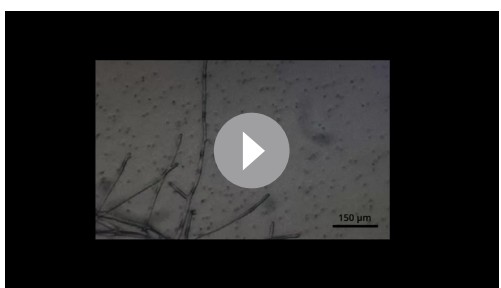

**Video 6.** Each hypha was dissected using laser microdissection and collected separately.
https://elifesciences.org/articles/107043/figures#video6

and Loewith, 2006; Kim and Guan, 2019). In both yeast and mammalian models, amino acids like branched-chain amino acids, glutamine, and alanine are crucial for nutrient sensing, forming the basis for TOR signaling activation (*Kim and Guan, 2019*; *Saxton and Sabatini, 2017*; *Meng et al., 2020*). These amino acids are like those contributing to nuclear increase in the RIB915 strain (*Figure 3G*). Furthermore, TOR inhibition by rapamycin in the RIB40 strain suppressed the formation of hyphae with increased nuclei (*Figure 3I–K*). Consistent with the activation of translation by the TOR pathway, the expression of ribosome-related genes was elevated in thicker hyphae . The TOR pathway directly regulates mitochondrial activity (*Düvel et al., 2010*). Thick hyphae exhibited heightened mitochondrial density and activity , suggesting that the TOR pathway supports the high ATP demand associated with rapid biosynthesis.

Cell volume and nuclear number mutually regulate each other, meaning that thicker hyphae always possess more nuclei. For the formation of thicker hyphae with numerous nuclei, an increase in cell volume and a corresponding increase in nuclei must occur simultaneously (*Figure 6A*). In our model, cell wall loosening at a branching site and regulation of cell volume by turgor pressure constitute necessary conditions for increasing cell volume and maintaining thick hyphae. RseA and MsyA may be involved in these processes. At the same time, enhanced translational capacity, possibly due to increased expression of ribosomal genes associated with TOR activation, and mechanisms that accelerate the cell cycle represent another essential condition that enables an increase in nuclear number. Both genetic potential and nutritional environmental signals are likely required for the formation of thick hyphae with a high number of nuclei. In the *A. oryzae* RIB915 strain, the number of nuclei and cell volume increased in response to specific amino acids, whereas in the *A. oryzae* RIB40 and other strains, both increased even without the addition of such amino acids, suggesting that they possess the genetic potential to respond independently of nutritional signals. When thick hyphae were cultured on fresh medium, thin hyphae initially emerged, suggesting the necessity of sustained high metabolic activity. Weakening the cell wall by treatment with a low concentration of calcofluor white did not lead to hyphal thickening or an increase in nuclear number. On the contrary, thick hyphae have thicker cell walls (*Figure 2H–K*), which are necessary to maintain the increased cell volume.

*A. nidulans* and *A. flavus* did not exhibit an increase in nuclear number. *A. sojae* displayed increased nuclear numbers, whereas *A. kawachii* and *A. luchuensis* did not (*Figure 1*). The 20 strains of *A. oryzae* selected from different clades showed an increase in nuclear number in the minimal medium supplemented with yeast extract. In the medium without yeast extract, there were strains that showed an increase in nuclear number and others that did not. These differences might be due to variations in their applications or the selection of strains with diverse characteristics, such as not only enzyme activity but also their impact on the flavor, aroma, and color of the final products. Additionally, there might be dominant mutations that activate the TOR pathway even in the minimal medium. Multiple gene mutations were identified as contributing to the phenotype of increased hyphal volume and nuclear number, suggesting that the phenotype was not the result of a single mutation but rather the gradual accumulation of mutations through breeding. The koji-fungus strains used to produce high-quality sake and soy sauce have been selectively bred over many years, long before the advent of modern biotechnology, to enhance their saccharolytic and proteolytic activities. Modern biotechnological bred strains such as *T. reesei* QM9414 and *P. chrysogenum* IFO4688 also exhibit a phenotype of thicker hyphae with increased nuclear numbers under nutrient-rich conditions (*Figure 5*). This consistency suggests that by breeding naturally multinucleate filamentous fungi under nutrient-rich conditions, it is possible to obtain strains with increased nuclear numbers and enhanced enzyme or compound production.

A previous study reported an increase in the number of nuclei in *A. nidulans* (*Clutterbuck, 1970*; *Vinck et al., 2005*). Here, we examined the nuclear distribution of *A. nidulans* grown on the culture

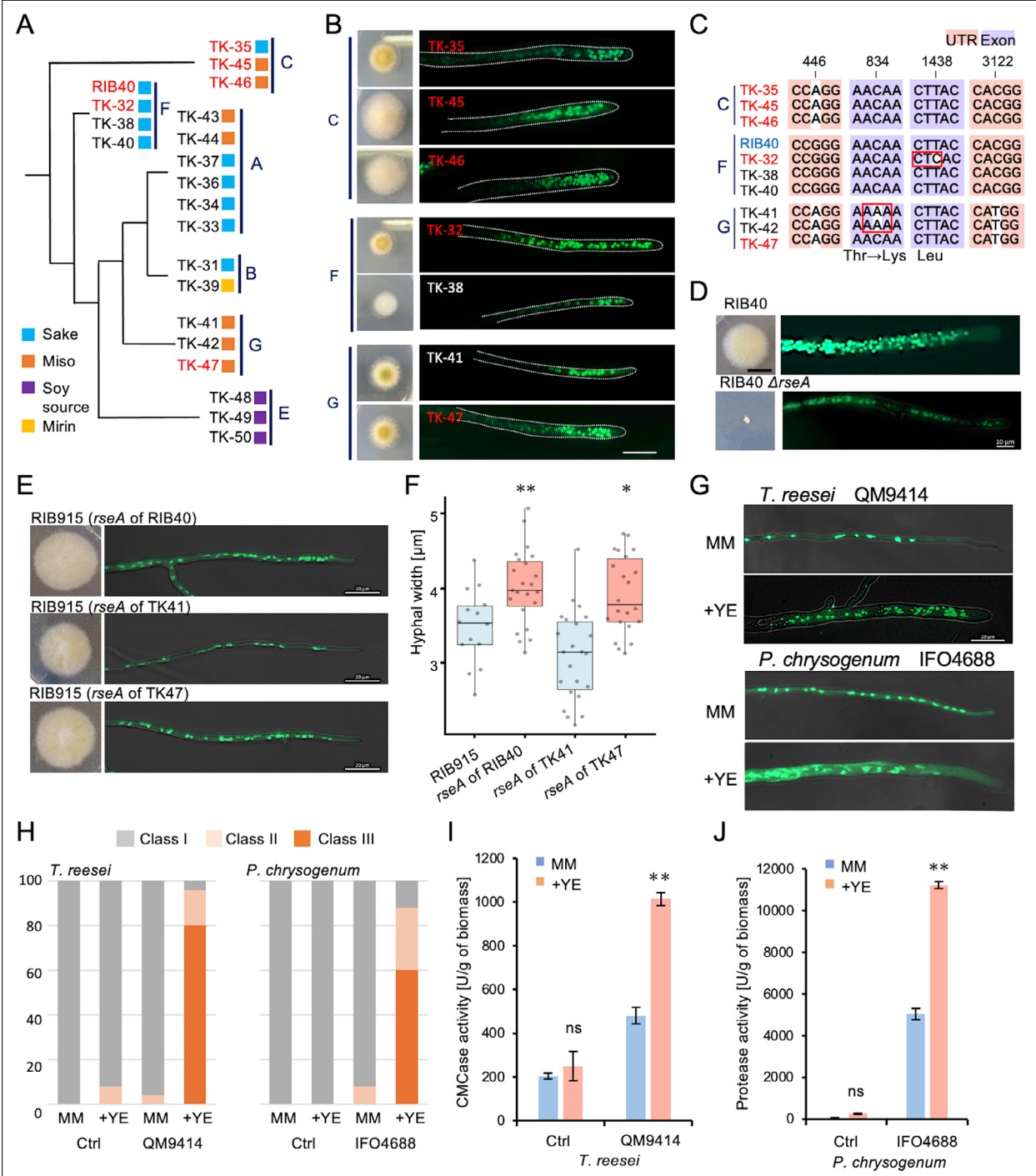

**Figure 5.** Comparison of industrial bred strains in *A. oryzae, T. reesei*, and *P. chrysogenum*. (**A**) Phylogenetic clades A–F of *A. oryzae* TK strains. The strains with increased nuclei are indicated in red. (**B**) Colonies and hyphae stained with SYBR Green of strains in clades C, F, and G grown for 3 days on the minimal medium. Scale bar: 20 μm. (**C**) Sequence differences in the *rseA* gene among strains in clades C, F, and G. UTR regions are marked in pink, exons in purple, and nucleotide differences in white. (**D**) Colonies and hyphae stained with SYBR Green of RIB40 and the Δ*rseA* strain grown for 3 days on the minimal medium. Scale bar: 10 μm. (**E**) Colonies and hyphae stained with SYBR Green of RIB915 expressing *rseA* gene from RIB40 and TK-47 strains with increased nuclei, and TK-41 strain without increased nuclei. Scale bar: 20 μm. (**F**) Hyphal width of RIB915 and strains in E (mean ± S.E., n=14–23, ** p < 0.01, * p < 0.05, t-test). Strains with *rseA* from strains with increased nuclei are marked in red, and those without are marked in blue. (**G**) Hyphal images stained with SYBR Green of *T. reesei* (QM9414) and *P. chrysogenum* (IFO4688) grown for 3 days on the minimal medium with or without 1% yeast extract. Scale bar: 20 μm. (**H**) Ratio of class I–III hyphae in the *T. reesei* and *P. chrysogenum* control and bred strains cultured on the minimal medium with or without 1% yeast extract for 3 days (*n* = 50). (**I**) CMCase activity per biomass of *T. reesei* industrial and control strains under the conditions in G

*Figure 5 continued on next page*

*Figure 5 continued*

(mean ± SE, *n* = 3, **p < 0.01, *t*-test). (**J**) Protease activity per biomass of *P. chrysogenum* industrial and control strains under the conditions in G (mean ± SE, *n* = 3, **p < 0.01, *t*-test).

The online version of this article includes the following figure supplement(s) for figure 5:

**Figure supplement 1.** *A. oryzae* TK strains phenotypes, rseA gene variants, and ectopic expression.

**Figure supplement 2.** Phenotype of industrial control strains.

media; however, we did not find class III hyphae as observed in *A. oryzae*. Even in *A. nidulans*, conidiophore stalks contain a high number of nuclei. It has been shown that *A. oryzae* has a taller conidiophore stalk (*Wada et al., 2012*). In the thick hyphae of *A. oryzae*, the expression level of *flbA*, an early regulator of conidiophore development (*Lee and Adams, 1994*), was elevated. This suggests that differentiation to aerial hyphae may be involved in the increase of hyphal volume and nuclear number.

We investigated whether the reduction in nuclear size observed in thick hyphae was due to a change from diploid to haploid status. However, no difference in GFP-histone fluorescence intensity was detected between thick and thin hyphae (*Figure 5—figure supplement 2D*). In both RIB40 and RIB915 strains, no significant difference in conidial size was observed despite the large difference in the number of nuclei within the hyphae (*Figure 5—figure supplement 2E*). These results suggest that both thick and thin hyphae remain haploid, and that the smaller nuclear size observed in thick hyphae

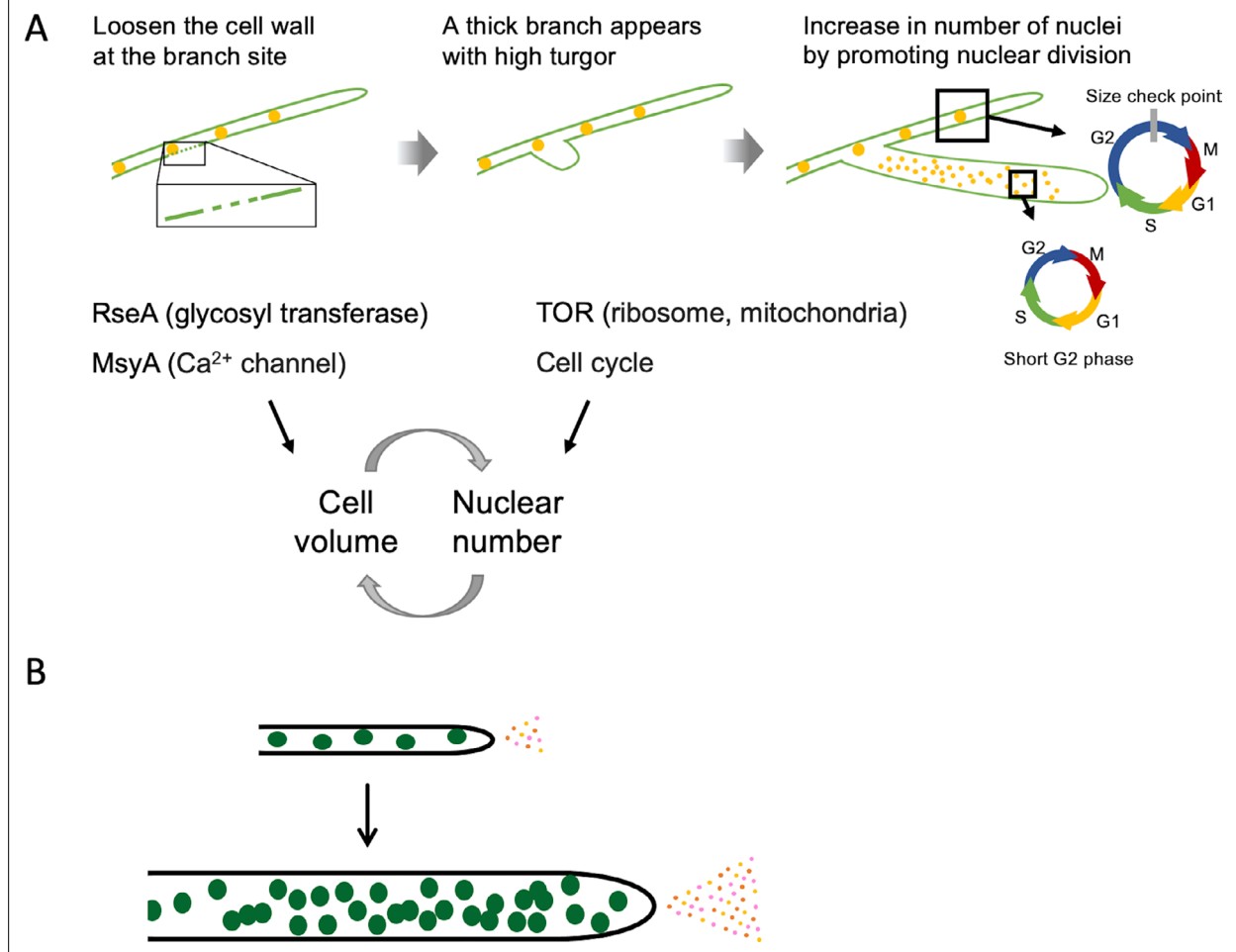

**Figure 6.** Working model of hyphal cell volume, nuclear number increase, and enhanced enzyme productivity. (**A**) Molecular mechanism by which cell volume and nuclear number regulate each other and each increases simultaneously. (**B**) An increase in the number of nuclei per hyphal cell enhances transcription, translation, and enzyme secretion per cell.

is likely due to a higher nuclear density. This is likely related to the phenomenon in which a decrease in cell size is accompanied by a reduction in nuclear size (*Lemière et al., 2022*).

The number of nuclei in filamentous fungi varies greatly among species (*Roper et al., 2011*). For example, *Neurospora crassa* hyphal cells can exceed 100 nuclei, which correlates with its exceptionally rapid growth rate and high protein secretion capacity (*Havlik et al., 2017*). An increase in the number of nuclei per cell is expected to enhance transcription and translation per cell, thereby improving enzyme production and secretion capacity (*Figure 6B*). Whereas maintaining a high nuclear number requires a substantial amount of energy, and in conditions where external nutrient supply is insufficient, cells rapidly die. Thus, the tendency to maintain a high nuclear number could be a selective pressure unlikely to occur in nature. The regulation of nuclear number and its ecological strategy are intriguing in other fungi such as *N. crassa*, which rapidly spreads after wildfires (*Jacobson et al., 2004*), and arbuscular mycorrhiza fungi that form symbiotic relationships with plants and contain thousands of nuclei within hyphae lacking septa (*Kokkoris et al., 2020*). In any case, koji-fungus strains were bred specifically for high enzyme production in nutrient-rich environments, and we propose that some of them acquired the specialized trait of increased cell volume and nuclear number through selection. This study elucidates the evolutionary processes driven by breeding selection in *A. oryzae* and the underlying mechanisms contributing to high enzyme productivity. These findings not only offer new approaches for optimizing filamentous fungi for bioindustry applications but also lead to the elucidation of the molecular mechanisms regulating cell volume and nuclear number in multinucleated organisms.

## Methods

### Strains and culture conditions

The filamentous fungal strains used in this study are listed in *Supplementary file 6*. *A. oryzae* RIB40 expressing H2B-GFP, UtH2BG stain, is described previously (*Yasui et al., 2020*). Minimal medium for *Aspergillus* species was shown in *Supplementary file 7*. When peptone, casamino acids, or yeast extract was added, the final concentration was 1%. Amino acids were added at a final concentration of 0.1%, B vitamins (biotin, pyridoxine hydrochloride, thiamin hydrochloride, riboflavin, PABA, and nicotinic acid) were added at 100 μM, and nucleotides (inosine and uridine) were added at 10 mM.

### Genetic manipulation

To delete target genes, genome editing plasmids and circular donor DNA plasmids were constructed as described previously (*Katayama et al., 2019*). The sgRNA cassette was designed under the U6 promoter with the following target sequences: GTGACGATCTACGGTAACGC for the *msyA* gene, GATCAGTTGAGCGTCCCCTA for the *msyB* gene, and AGCTAGCGCAGCCGGTCTGC for the *rseA* gene. The genome editing plasmid was constructed by introducing the sgRNA expression cassette into the SmaI restriction site of the pRGE-gRT6 plasmid (*Katayama et al., 2019*). For the donor plasmid, 1 kb fragments upstream and downstream of the target genes were amplified, ligated, and inserted into the linearized pUC19 vector (Takara Bio). Both plasmids were applied for transformation using the *A. oryzae* RIB40 strain. To amplify the *rseA* region, spanning 1 kb upstream to 1 kb downstream, the templates used were *A. oryzae* RIB40, TK-42, and TK-47 strains. The amplified fragments were inserted into the HindIII site of the pPTRI vector (TaKaRa) to construct the plasmid. Transformations were performed using the protoplast-PEG method according to the manufacturer's protocol for pPTRI. Transformants were selected on the minimal medium plates containing 0.1 mg/l pyrithiamine. Integration of constructs was confirmed by PCR analysis.

### Microscopy

Images were captured using an Axio Observer Z1 inverted epifluorescence microscope (Carl Zeiss) equipped with Plan-Apochromat objectives, an AxioCam 506 mono camera, and a Colibri.2 LED light source. The stage temperature was maintained at 30°C with a thermoplate (Tokai Hit). Nuclei were stained with SYBR Green (Takara), mitochondria with Rhodamine 123 (Invitrogen), the cell membrane with FM4-64 (Invitrogen). Images were collected using Zen (Carl Zeiss) and ImageJ software.

### Image analysis

The distribution pattern of nuclei was classified into the following three types: class I: nuclei distribute at a constant interval without overlapping; class II: nuclei align but sometimes overlap; class III: nuclei

scattered throughout hyphae but not aligned. Nuclei, hyphal size, and fluorescence intensity were quantified using Zen and ImageJ software. Z-stack 3D reconstruction and nuclear color coding were performed with Imaris (Oxford Instruments). Hyphal elongation distances were analyzed using the MTrackJ plugin in ImageJ. For *Figure 2E*, nuclei were binarized, and those in non-increased hyphae were displayed in white based on size and circularity.

## Transmission electron microscopy

The culture samples were embedded in 1% water agar and fixed with 2.5% glutaraldehyde in 0.1 M phosphate buffer (pH 7.0) at room temperature for 2 hr. Sample preparation and sectioning were performed at the Institute of Medicine, University of Tsukuba. Observations were conducted using a JEM-1400 transmission electron microscope (JEOL).

## Imaging with microfluidic devices

Microfluidic devices were fabricated as described previously (*Yasui et al., 2020*). Two types of devices were constructed: a 2D observation device to visualize hyphal expansion (*Figure 2—figure supplement 1A*) and a single-hypha isolation device for enzymatic activity measurement (*Figure 3—figure supplement 1A*). Conidia suspensions were loaded into a 10-ml plastic syringe (SS-10ESZ, Terumo) and connected to polyethylene tubing (inner diameter 0.38 mm, outer diameter 1.09 mm, BD Intramedic). Air was expelled from the syringe, and the suspension was injected into the device inlet. The number of conidia flowing through the microfluidic channel was adjusted under a microscope. The syringe was then replaced with another containing medium, and the medium was delivered at the minimum flow rate using a syringe pump (YSP-101, YMC). The device was incubated at 30°C on a thermal stage (TOKAI HIT) and imaged using an Axio Observer Z1 microscope (Carl Zeiss). α-Amylase activity in the device was measured using the EnzChek Ultra amylase assay kit (Invitrogen).

## Enzyme activity assays

Fungal conidia $10^4$ were inoculated onto 25 ml of medium in a petri dish and statically cultured at 30°C for 4 days. Protein concentration and enzymatic activities in the culture supernatant were measured. Protein quantification was performed using the Bradford assay (Protein Assay Kit, Bio-Rad), with bovine gamma globulin as the standard. Carboxymethyl cellulase (CMCase) activity was determined by measuring reducing sugars using the 3,5-dinitrosalicylic acid method. CMCase activity was assayed at 50°C for 15 min in a reaction mixture containing 50 mM sodium acetate buffer (pH 5.0) and 1% carboxymethyl cellulose. One unit of enzyme activity was defined as the amount of enzyme required to release 1 μmol of reducing sugar (as glucose equivalent) per minute. α-Amylase activity was measured using an α-Amylase Assay Kit (Kikkoman Biochemifa) according to the manufacturer's instructions. Acid carboxypeptidase activity was determined using the Acid Carboxypeptidase Assay Kit (Kikkoman Biochemifa) following the provided protocol.

## RNA-seq analysis

Sterilized cellophane was placed on agar plates, and conidia were inoculated. After culturing at 30°C for 3 days, samples were frozen, homogenized with a mortar and pestle, and total RNA was extracted using the RNeasy Mini Kit (QIAGEN). Library preparation, sequencing, and partial data analysis were performed by the Faculty of Medicine, University of Tsukuba. Sequencing was conducted using 36 bp paired-end reads on the Illumina NextSeq 500, and reads were mapped to the *A. oryzae* RIB40 reference genome using the CLC Genomic Workbench (QIAGEN). For laser microdissection, sterilized membrane slides (Carl Zeiss) were immersed in MM liquid, and conidia were inoculated onto the membrane. After static culturing at 30°C for 3 days, sections were collected using a PALM MicroBeam laser microdissection system (Carl Zeiss). RNA was extracted from 500 to 700 sections, respectively, using the RNeasy Mini Kit. Library preparation and sequencing were performed by Takara Bio Inc, using 150 bp paired-end reads on the Illumina NovaSeq 6000. Reads were mapped to the *A. oryzae* RIB40 reference genome using the CLC Genomic Workbench. Genes showing a ≥4-fold change in expression were considered significant, and GO enrichment analysis of biological processes was performed using ShinyGO 0.81 (*Ge et al., 2020*). Heatmaps were visualized using the Python Seaborn package (*Waskom, 2021*). Gene annotations were obtained from FungiDB.

### Low osmotic pressure shock analysis

Conidia were spot inoculated onto the minimal medium containing 1 M NaCl and cultured at 30°C for 3 days. Colony tips were treated with distilled water for 3 min, and the number of tip ruptures was counted. Hyphal tip swelling was assessed by staining hyphae cultured in the minimal medium liquid for 3 days with FM4-64, mounting them on a coverslip with a small volume of medium, and adding an equal volume of distilled water.

### Raman spectroscopy

The conidia were inoculated into minimal medium liquid in a glass-bottom dish and cultured at 30°C for 3 days. Raman spectra were acquired using an XploRA PLUS Raman microscope, as described previously (*Maini Rekdal et al., 2024a*). Hyphae attached to the dish bottom were irradiated with a 532-nm laser at 0.93 mW for 30 s. Spike artifacts caused by cosmic rays were manually removed. Ten spectra were averaged, and background signals attributed to the medium and glass were subtracted.

### Protein structure analysis

RseA protein structure was predicted using AlphaFold2 and visualized with PyMOL (v1.20). Domains were annotated using InterPro.

### Atomic force microscopy

The conidia were inoculated into minimal medium liquid on a Poly-L-Lysine-coated (Sigma-Aldrich) glass-bottom dish (ibidi) and cultured at 30°C for 3 days. We used a JPK Nanowizard 4 (Bruker) equipped with an inverted fluorescence microscope (Eclipse Ti2, Nikon). The temperature was maintained at 30°C using a dish heater integrated into the stage. BL-AC40TS-C2 cantilevers (Olympus, spring constant approximately 0.1 N/m) were used for AFM imaging under the following conditions: QI mode, $1.5 \times 1.5$ μm or $0.5 \times 0.5$ μm scan size, $128 \times 128$ pixels, Z-length 1 μm, setpoint 0.1 nN, and Z speed 166 μm/s. The Young's modulus was calculated using JPKSPM Data Processing software (Bruker). After averaging and baseline subtraction, Young's modulus was calculated using a Hertz model with a triangular pyramid half-cone angle of 35°.

### SNP analysis

The genome data of *A. oryzae* required for SNP identification was obtained from the National Center for Biotechnology Information (NCBI) and mapped to *A. oryzae* RIB40 reference sequence using BWA-MEM2 (*Vasimuddin et al., 2019*). The SNPs were listed by FreeBayes (*Garrison and Marth, 2012*), and detailed comparative analysis of genomes was performed on Integrative Genome Viewer (IGV) manually.

### RNA-seq

The RNA-seq data have been deposited in DDBJ as BioProject PRJDB19992.

## Acknowledgements

This work was funded by MEXT KAKENHI grant numbers 21H02095, 21K19062, 25K01927 and Scientific Research on Innovative Areas 'Post-Koch Ecology' grant number 22H04878 to NT. Ohsumi Frontier Science Foundation, Noda Institute for Scientific Research Grant, and Japan Science and Technology Agency (JST) ERATO grant number JPMJER1502 to NT. AFM measurement was supported by the World Premier International Research Center Initiative (WPI).

## Additional information

#### Competing interests

Hideyuki Yamashita: employee of Higuchi Matsunosuke Shoten Co., Ltd. The other authors declare that no competing interests exist.

## Funding

| Funder | Grant reference number | Author |
|---|---|---|
| Ministry of Education, Culture, Sports, Science and Technology | 21H02095 | Norio Takeshita |
| Ministry of Education, Culture, Sports, Science and Technology | 21K19062 | Norio Takeshita |
| Ministry of Education, Culture, Sports, Science and Technology | 25K01927 | Norio Takeshita |
| Ministry of Education, Culture, Sports, Science and Technology | 22H04878 | Norio Takeshita |
| Ohsumi Frontier Science Foundation | | Norio Takeshita |
| Noda Institute for Scientific Research | | Norio Takeshita |
| Japan Science and Technology Agency | 10.52926/jpmjer1502 | Norio Takeshita |

The funders had no role in study design, data collection, and interpretation, or the decision to submit the work for publication.

## Author contributions

Ayaka Itani, Data curation, Formal analysis, Validation, Investigation, Visualization, Methodology, Writing – original draft; Haruto Motomura, Formal analysis, Validation, Investigation; Ken Oda, Hideyuki Yamashita, Ken-ichi Kusumoto, Resources; Kanae Sakai, Formal analysis; Shinsuke Shigeto, Takehiko Ichikawa, Hosain Mohammad Mubarak, Takeshi Fukuma, Methodology; Takuya Katayama, Shunsuke Masuo, Resources, Methodology; Jun-ichi Maruyama, Resources, Supervision, Methodology; Naoki Takaya, Conceptualization, Resources, Project administration; Norio Takeshita, Conceptualization, Supervision, Funding acquisition, Writing – original draft, Project administration, Writing – review and editing

## Author ORCIDs

Shinsuke Shigeto ⓘ https://orcid.org/0000-0002-2035-2068
Shunsuke Masuo ⓘ https://orcid.org/0000-0002-0454-3315
Norio Takeshita ⓘ https://orcid.org/0000-0003-2666-4991

Reviewer #1 (Public review): https://doi.org/10.7554/eLife.107043.4.sa1
Reviewer #2 (Public review): https://doi.org/10.7554/eLife.107043.4.sa2
Reviewer #3 (Public review): https://doi.org/10.7554/eLife.107043.4.sa3
Author response https://doi.org/10.7554/eLife.107043.4.sa4

# Additional files

## Supplementary files

Supplementary file 1. Annotated data of RNA-seq in *A. oryzae* RIB40 and RIB915, and *A. nidulans* grown in the minimal medium with or without yeast extract.

Supplementary file 2. Annotated data of RNA-seq in *A. oryzae* RIB40 thick or thin hyphae.

Supplementary file 3. Annotated data of RNA-seq common in upregulated gene in *A. oryzae* RIB915 grown with yeast extract and in *A. oryzae* RIB40 thick hyphae.

Supplementary file 4. SNP analysis of ORFs in clade F between TK-32 and TK-38.

Supplementary file 5. SNP analysis of ORFs in clade G between TK-41 and TK-47.

Supplementary file 6. Strains used in this study.

Supplementary file 7. Composition of minimal medium.
MDAR checklist

### Data availability
The RNA-seq data have been deposited in DDBJ as BioProject PRJDB19992.

The following dataset was generated:

| Author(s) | Year | Dataset title | Dataset URL | Database and Identifier |
|---|---|---|---|---|
| Itani A | 2025 | The increase in cell volume and number of nuclei of the Koji-kin Aspergillus oryzae contributes to its high enzyme productivity | https://ddbj.nig.ac.jp/search/entry/bioproject/PRJDB19992 | DNA Data Bank of Japan, PRJDB19992 |

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
