## [Editor Report · eLife Assessment]

The ratio of nuclei to cell volume is a well-controlled parameter in eukaryotic cells. This study now reports **important** findings that expand our understanding of the regulatory relationship between cell size and number of nuclei. The evidence supporting the conclusions is **convincing** obtained by applying appropriate and validated methodology in line with current state-of-the-art. The paper will be of broad interest for cell biologists and fungal biotechnologists seeking to understand mechanisms determining cell size and number of nuclei and why this knowledge might also be of importance for the production of enzymes and thus production strains not only of Aspergillus oryzae but also other industrially used fungi.

---

## [Referee Report · Reviewer #1 (Public review)]

Filamentous fungi are established work horses in biotechnology with *Aspergillus oryzae* as a prominent example with a thousand-year of history. Still the cell biology and biochemical properties of the production strains is not well understood. The paper of the Takeshita group describes the change in nuclear numbers and correlate it to different production capacities. They used microfluidic devices to really correlate the production with nuclear numbers. In addition, they used microdissection to understand expression profile changes and found an increase of ribosomes. The analysis of two genes involved in cell volume control in *S. pombe* did not reveal conclusive answers to explain the phenomenon. It appears that it is a multi-trait phenotype. Finally, they identified SNPs in many industrial strains and tried to correlate them to the capability of increasing their nuclear numbers.

The methods used in the paper range from high quality cell biology, Raman spectroscopy to atomic force and electron microscopy and from laser microdissection to the use of microfluidic devices to study individual hyphae.

This is a very interesting, biotechnologically relevant paper with the application of excellent cell biology.

Comments on revised version:

The authors addressed all suggestions satisfactorily.

---

## [Referee Report · Reviewer #2 (Public review)]

Summary:

In the study presented by Itani and colleagues it is shown that some strains of Aspergillus oryzae - especially those used industrially for the production of sake and soy sauce - develop hyphae with a significantly increased number of nuclei and cell volume over time. These thick hyphae are formed by branching from normal hyphae and grow faster and therefore dominate the colonies. The number of nuclei positively correlates with the thicker hyphae and also the amount of secreted enzymes. The addition of nutrients such as yeast extract or certain amino acids enhanced this effect. Genome and transcriptome analyses identified genes, including rseA, that are associated with the increased number of nuclei and enzyme production. The authors conclude from their data involvement of glycosyltransferases, calcium channels and the tor regulatory cascade in regulation of cell volume and number of nuclei. Thicker hyphae and an increased number of nuclei was also observed in high-production strains of other industrially used fungi such as Trichoderma reesei and Penicillium chrysogenum, leading to the hypothesis that the mentioned phenotypes are characteristic of production strains which is of significant interest for fungal biotechnology.

Strengths:

The study is very comprehensive and involves application of divers state-of-the-art cell biological, biochemical and genetical methods. Overall, the data are properly controlled and analyzed, figures and movies are of excellent quality.

The results are particularly interesting with regard to the elucidation of molecular mechanisms that regulate the size of fungal hyphae and their number of nuclei. For this, the authors have discovered a very good model: (regular) strains with a low number of nuclei and strains with high number of nuclei. Also, the results can be expected to be of interest for the further optimization of industrially relevant filamentous fungi.

In the revision the authors addressed all my comments and as a result produced an even stronger study.

---

## [Referee Report · Reviewer #3 (Public review)]

Summary:

The authors seek to determine the underlying traits that support the exceptional capacity of Aspergillus oryzae to secrete enzymes and heterologous proteins. To do so, they leverage the availability of multiple domesticated isolates of A. oryzae along with other Aspergillus species to perform comparative imaging and genomic analysis.

Strengths:

The strength of this study lies in the use of multifaceted approaches to identify significant differences in hyphal morphology that correlate with enzyme secretion, which is then followed by the use of genomics to identify candidate functions that underlie these differences.

Weaknesses:

Although the image analysis and data interpretation is convincing, the genetic data supporting the author's model is somewhat more speculative and will likely require additional investigation.

Overall, the authors have achieved their aims in that they are able to clearly document the presence of two distinct hyphal forms in A. oryzae and other Aspergillus species, and to correlate the presence of the thicker rapidly growing form with enhanced enzyme secretion. The image analysis is convincing. The discovery that addition of yeast extract and specific amino acids can stimulate formation of the novel hyphal form is also notable. Although the conclusions are generally supported by the results, this is perhaps less so for the genetic analysis as it remains unclear how direct the role of RseA and the calcium transporters might be in supporting the formation of the thicker hyphae.

The results presented here will impact the field. The complexity of hyphal morphology and how it affects secretion are not well understood despite the importance of these processes for the fungal lifestyle. In addition, the description of approaches that can be used to facilitate the study of these different hyphal forms (i.e., stimulation using yeast extract or specific animo acids) will benefit future efforts to understand the molecular basis of their formation.

---

## [Author Response]

The following is the authors’ response to the original reviews.

**Reviewer #1 (Public review):**
Filamentous fungi are established workhorses in biotechnology, with *Aspergillus oryzae* as a prominent example with a thousand-year history. Still, the cell biology and biochemical properties of the production strains is not well understood. The paper of the Takeshita group describes the change in nuclear numbers and correlates it to different production capacities. They used microfluidic devices to really correlate the production with nuclear numbers. In addition, they used microdissection to understand expression profile changes and found an increase in ribosomes. The analysis of two genes involved in cell volume control in *S. pombe* did not reveal conclusive answers to explain the phenomenon. It appears that it is a multi-trait phenotype. Finally, they identified SNPs in many industrial strains and tried to correlate them to the capability of increasing their nuclear numbers.The methods used in the paper range from high-quality cell biology, Raman spectroscopy, to atomic force and electron microscopy, and from laser microdissection to the use of microfluidic devices tostudy individual hyphae.This is a very interesting, biotechnologically relevant paper with the application of excellent cell biology. I have only minor suggestions for improvement.

We sincerely appreciate your fair and positive evaluation of our work. Thank you for your suggestions for improvement. We respond to each of them appropriately.

**Reviewer #2 (Public review):**
Summary:In the study presented by Itani and colleagues, it is shown that some strains of Aspergillus oryzae - especially those used industrially for the production of sake and soy sauce - develop hyphae with a significantly increased number of nuclei and cell volume over time. These thick hyphae are formed by branching from normal hyphae and grow faster and therefore dominate the colonies. The number of nuclei positively correlates with the thicker hyphae and also the amount of secreted enzymes. The addition of nutrients such as yeast extract or certain amino acids enhanced this effect. Genome and transcriptome analyses identified genes, including rseA, that are associated with the increased number of nuclei and enzyme production. The authors conclude from their data involvement of glycosyltransferases, calcium channels, and the tor regulatory cascade in the regulation of cell volume and number of nuclei. Thicker hyphae and an increased number of nuclei were also observed in high-production strains of other industrially used fungi such as Trichoderma reesei and Penicillium chrysogenum, leading to the hypothesis that the mentioned phenotypes are characteristic of production strains, which is of significant interest for fungal biotechnology.Strengths:The study is very comprehensive and involves the application of diverse state-of-the-art cell biological, biochemical, and genetic methods. Overall, the data are properly controlled and analyzed, figures andmovies are of excellent quality.The results are particularly interesting with regard to the elucidation of molecular mechanisms that regulate the size of fungal hyphae and their number of nuclei. For this, the authors have discovered a very good model: (regular) strains with a low number of nuclei and strains with a high number of nuclei. Also, the results can be expected to be of interest for the further optimization of industrially relevant filamentousfungi.Weaknesses:There are only a few open questions concerning the activity of the many nuclei in production strains (active versus inactive), their number of chromosomes (haploid/diploid), and whether hyper-branching always leads to propagation of nuclei.

We are very grateful for your recognition of our findings, the proposed model, and their significance for future applications. We are grateful for the questions, which contribute to a more accurate understanding.

Our responses to each are provided below.

**Reviewer #3 (Public review):**
Summary:The authors seek to determine the underlying traits that support the exceptional capacity of Aspergillus oryzae to secrete enzymes and heterologous proteins. To do so, they leverage the availability of multiple domesticated isolates of A. oryzae along with other Aspergillus species to perform comparative imaging and genomic analysis.Strengths:The strength of this study lies in the use of multifaceted approaches to identify significant differences in hyphal morphology that correlate with enzyme secretion, which is then followed by the use of genomics to identify candidate functions that underlie these differences.Weaknesses:There are aspects of the methods that would benefit from the inclusion of more detail on how experiments were performed and data interpreted.Overall, the authors have achieved their aims in that they are able to clearly document the presence of two distinct hyphal forms in A. oryzae and other Aspergillus species, and to correlate the presence of the thicker, rapidly growing form with enhanced enzyme secretion. The image analysis is convincing. The discovery that the addition of yeast extract and specific amino acids can stimulate the formation of the novel hyphal form is also notable. Although the conclusions are generally supported by the results, this is perhaps less so for the genetic analysis as it remains unclear how direct the role of RseA and the calcium transporters might be in supporting the formation of the thicker hyphae.The results presented here will impact the field. The complexity of hyphal morphology and how it affects secretion is not well understood despite the importance of these processes for the fungal lifestyle. In addition, the description of approaches that can be used to facilitate the study of these different hyphal forms (i.e., stimulation using yeast extract or specific amino acids) will benefit future efforts to understand the molecular basis of their formation.

We are very grateful for your fair and thoughtful evaluation of our work. We agree that the genetic analysis in the latter part is relatively weaker compared to the imaging analysis in the first half. Rather than a single mutation causing a dramatic phenotypic change, we believe that the accumulation of various mutations through breeding leads to the observed phenotype, making it difficult to clearly demonstrate causality. Since transcriptome and SNP analyses have revealed key pathways and phenotypes, it would be gratifying if these insights could contribute to future applications utilizing filamentous fungi.

**Reviewer #1 (Recommendations for the authors):**
I was wondering what happens if thick hyphae were taken as inoculum for a new colony or thin hyphae. Is it possible to enrich for one or the other type of hyphae? Perhaps in the presence of yeast extract or certain amino acids.

Added an explanation in the discussion.

L304-306. When thick hyphae were cultured on fresh medium, thin hyphae initially emerged, suggesting that sustained metabolic activity is required for the formation of thick hyphae with a high number of nuclei.

L120-121. In some cases, thick hyphae emerged by branching from thick hyphae (Fig. 2D, left), while in other cases, thin hyphae emerged from thick hyphae (Fig. 2D, right). Thin hyphae emerge in the early stage of cultivation even in the presence of yeast extract or certain amino acids.

In the Discussion, they hypothesize that the primary effect could be on cell wall rigidity. I am wondering if that hypothesis could be tested by adding, for instance, sublethal concentrations of cytochalasin to hyphae of A. nidulans to weaken the cell wall.

The question is reasonable. To ensure accurate understanding, we moved Fig. S6 to Fig. 6 and revised the discussion as follows.

L294-295. In our model, cell wall loosening at a branching site and regulation of cell volume by turgor pressure constitute necessary conditions for increasing cell volume and maintaining thick hyphae. L306-309. Weakening the cell wall by treatment with a low concentration of calcofluor white did not lead to hyphal thickening or an increase in nuclear number. On the contrary, thick hyphae have thicker cell walls (Fig. 2H-K), which are necessary to maintain the increased cell volume.

I recommend including some older literature. It was described already 20 years ago that A. nigerdifferentiates hyphae with different capacities to secrete proteins (PMID: 16238620). In addition, there are old reports in A. nidulans reporting high numbers of nuclei (https://doi.org/10.1099/00221287-60-1-133). Perhaps it is worth trying to reproduce those cultural conditions. At least this should be discussed. In the same line, the number of nuclei increases a lot in the stalk of conidiophores in A. nidulans. These observations could be used as examples that the phenomenon observed in A. oryzae may be of general importance.

Thank you for the suggestion. It is a very interesting proposal. We checked the nuclei distribution of A. nidulans on the media and added the following discussion.

L328-334. A previous study reported an increase in the number of nuclei in A. nidulans (62, 63). Here, we examined the nuclear distribution of A. nidulans grown on the culture media, however, did not find class III hyphae as observed in A. oryzae. Even in A. nidulans, conidiophore stalks contain a high number of nuclei. It has been shown that A. oryzae has a taller conidiophore stalk (64). In the thick hyphae of A. oryzae, the expression level of flbA, an early regulator of conidiophore development (65), was elevated. This suggests that differentiation to aerial hyphae may be involved in the increase of hyphal volume and nuclear number.

(62) Clutterbuck A.J. Synchronous Nuclear Division and Septation in Aspergillus nidulans. J Gen Microbiol 60, 133-135 (1970).

(63) Vinck, A., Terlou, M., Pestman, W.R., Martens, E.P., Ram, A.F., van den Hondel, C.A., Wösten, H.A. Hyphal differentiation in the exploring mycelium of Aspergillus niger. Mol Microbiol 58, 693-9 (2005).

(64) Wada R, Maruyama J, Yamaguchi H, Yamamoto N, Wagu Y, Paoletti M, Archer DB, Dyer PS, Kitamoto K. Presence and functionality of mating type genes in the supposedly asexual filamentous fungus Aspergillus oryzae. Appl Environ Microbiol 78, 2819-29 (2012).

(65) Lee, B.N., Adams, T.H. Overexpression of flbA, an early regulator of Aspergillus asexual sporulation, leads to activation of brlA and premature initiation of development. Mol Microbiol 14, 323-34 (1994).

**Reviewer #2 (Recommendations for the authors):**
I suggest addressing the following questions to strengthen the manuscript:(1) Do the authors have an explanation for their result that with an increase in the number of nuclei the individual nucleus is smaller? Have the authors checked whether all the nuclei are haploid or diploid?

Thank you for the very important question. We added new results to Fig. S5D and S5E and the following discussion.

L335-340. We investigated whether the reduction in nuclear size observed in thick hyphae was due to a change from diploid to haploid status. However, no difference in GFP-histone fluorescence intensity was detected between thick and thin hyphae (Fig. S5D). In both RIB40 and RIB915 strains, no significant difference in conidial spore size was observed despite the large difference in the number of nuclei within the hyphae (Fig. S5E). These results suggest that both thick and thin hyphae remain haploid, and that the smaller nuclear size observed in thick hyphae is likely due to a higher nuclear density.

(2) In this context, the biological relevance of the increase in the number of nuclei should also be discussed in more detail. It remains to be clarified whether in hyphae with a high number of nuclei all nuclei are functionally active or whether many nuclei are possibly "inactive". Studies on the transcriptional activity of individual nuclei or on DNA replication (e.g., by EdU labeling) could clarify this.

Added the explanation below.

L102-105. The transcriptional activity of each nucleus is unknown. However, a previous study (Yasui et al., FBB 2020) demonstrated that nuclear division is synchronized even when there are more than 200 nuclei. This suggests that DNA replication occurs similarly in most nuclei. Furthermore, since the germination rate of conidia and the colonies formed from individual conidia show no significant abnormalities, it is suggested that nearly all nuclei possess normal genomes and chromosomes.

(3) It becomes not entirely clear what the underlying signal is that causes a thin hypha to branch into a thick multinucleated cell. This needs to be discussed in more detail.

Thanks for the suggestion. We clarified the signal to increase nuclear number and cell volume.

L294-309. Although it is speculative, we propose a model to aid interpretation in the discussion. We have clarified that both genetic potential and environmental signals such as nutrients are important.

(4) Is increased branching always correlated with an increased number of nuclei?

It is not an increase in branching, but rather the thickening of hyphae and an increase in cell volume that is consistently associated with an increase in nuclear number. Approximately 40 hours after inoculation, within 400 μm from the tip, the number of branches was 3.4 (SD=2.4) in thin hyphae and 2.6 (SD=0.5) in thick hyphae, suggesting that branching does not increase (n=4). Since thick hyphae elongate faster, it seems that fewer branches are present near the tip, even if the branching frequency itself remains unchanged.

(5) The abstract does not summarize the many findings of the manuscript in an adequate way.

abstract change

Minor:(1) Lines 49-50: Why italics?

corrected.

(2) Line 179: process.

corrected.

(3) Lines 313-314: Do not forget (and discuss) in this context mycorrhiza fungi with up to thousands of nuclei that were apparently selected during evolution for this high number of nuclei.

Thank you for the very interesting suggestion. We have added the following discussion.

L339-351. The regulation of nuclear number and its ecological strategy are intriguing in other fungi such as *N. crassa*, which rapidly spreads after wildfires (68), and arbuscular mycorrhiza fungi that form symbiotic relationships with plants and contain thousands of nuclei within hyphae lacking septa (69).

(68) Jacobson, D. J. et al. Neurospora in temperate forests of western North America. Mycologia 96, 66–74 (2004).

(69) Kokkoris V, Stefani F, Dalpé Y, Dettman J, Corradi N. Nuclear Dynamics in the Arbuscular Mycorrhizal Fungi. Trends Plant Sci. 25, 765-778 (2020).

(4) Lines 356-358: many typos.

corrected.

**Reviewer #3 (Recommendations for the authors):**

Specific suggestions or clarifications for the authors include:

(1) Lines 49-50: Is this sentence italicized for a reason?

It was a mistake, so we have corrected it.

(2) Line 83: More detail on the specific characteristics of the different classes of hyphae would be helpful. Perhaps include a schematic drawing that emphasizes the differences between class I,II, and III hyphae.

L398-400. The classification is described in the Methods section: Class I – nuclei are distributed at regular intervals without overlapping; Class II – nuclei are aligned but occasionally overlap; Class III – nuclei are scattered throughout the hyphae without alignment. Representative images are shown in a previous study (Yasui et al., FBB 2020).

L82-84. We have added this information to clarify the classification.

(3) Lines 102-103: It was not very clear how this experiment was done. Are you counting nuclei within 100 um of the tip? Are these all in one hyphal compartment? These details could be provided in a drawing that would make it easier for the reader to understand how this was done.

L109. Due to variation in the distance from the hyphal tip to the septum, we counted the number of nuclei within 100 μm from the hyphal tip. When septa were present, nuclei were counted in the same manner, so multiple compartments may be included. Changed the explanation.

(4) Lines 134-140: Is there a way to calibrate levels of secreted protein or amylase activity per nucleus? That is, if the ratio of cytoplasmic volume per nucleus is constant, does the same apply to the secreted product? Knowing this would help to clarify whether the key feature in enhanced secretion is nuclear (e.g., gene expression) versus a cytoplasmic trait (e.g., vesicle trafficking).

Enzyme activity was measured across the entire mycelium, which includes a mixture of hyphae with high and low numbers of nuclei. Therefore, it is difficult to assess the correlation between enzyme activity and nuclear number. Enzyme activity was normalized by fungal biomass. The size of each colony is shown in Fig. 1B. Additionally, the correlation between the proportion of hyphae with increased nuclear number and enzyme activity is shown in Fig. 3H. In the experiment where enzyme activity was measured in a single hypha, we attempted to measure the number of nuclei; however, we could not use the nuclear GFP strain because the substrate exhibits green fluorescence. DAPI staining also failed due to limited dye access to the microfluidic channel. Changed the section title, ‘Increase in nuclear number and enzyme secretion’ from ‘Correlation between nuclear number and enzyme secretion’.

(5) Line 151 and Figure 3F: YE also triggered a ~5-fold enhancement of secretion in A. nidulans without a concomitant increase in hyphal width. This merits some comment in the text.

Added an explanation, L156-157.

In A. nidulans, the addition of yeast extract did not cause a dramatic increase in nuclear number, but hyphal width increased by 1.4-times and protein secretion increased by 5.1-times.

(6) Line 252: Were nimE levels detected or altered in thick hyphae? The levels of this cycling might play a more important role in a shortened cell cycle than the authors have considered, especially as NimE functions during both G1 and G2.

Added an explanation below, L260-262.

The expression level of nimE (AO090003000993) was low in both thick and thin hyphae, with no significant difference observed. As known in other organisms, its function is likely regulated through phosphorylation and the protein degradation.

(7) Line 254: Please provide a citation for the statement that branches emerge as a result of cell wall loosening.

rephrased and added citation, L263.

Branching is thought to occur through the degradation and reconstruction of the cell wall at the branching site (54).

Harris SD. Branching of fungal hyphae: regulation, mechanisms and comparison with other branching systems. Mycologia 100, 823-32 (2008).

(8) Lines 275-277: It would be interesting to know whether the addition of rapamycin also suppressed the ability of amino acids to trigger greater numbers of class III hyphae.

We added new results at Fig. S2G.

L168. Rapamycin decreased the ratio of hyphae with increased nuclei even in the medium with yeast extract (Fig. S2G).

(9) Lines 282-289: My sense is that this model is too speculative at this time. The role of RseA seems very broad based on the strong deletion phenotype. How would the removal of RseA be regulated to limit its effect to the branch site? Also, the msyA deletion phenotype isn't entirely consistent with what you would expect if it were necessary to maintain thick hyphae. Lastly, the authors do not show that translational capacity is enhanced in thick hyphae. I would suggest that these statements be tempered to some degree.

Thank you for your comment. We agree that it was too speculative, whereas we believe that some explanatory interpretation is necessary. Therefore, we have revised the text as follows, L294-300. In our model, cell wall loosening during branching and regulation of cell volume by turgor pressure constitute necessary conditions for increasing cell volume and maintaining thick hyphae. RseA and MsyA may be involved in these processes. At the same time, enhanced translational capacity by increased expression of ribosomal genes, possibly due to associated with TOR activation by specific amino acids, and mechanisms that accelerate the cell cycle represent another essential condition that enables an increase in nuclear number.

(10) General: how do the authors reconcile the observation that YE and amino acids stimulate the formation of thicker hyphae, yet the time lapse imaging (Figure 2E) suggests that these hyphae arise at a later time during colony development when these resources might be limiting? The authors should consider providing some insight into this in the Discussion.

L300-305. Added a discussion below.

Both genetic potential and nutritional environmental signals are likely required for the formation of thick hyphae with a high number of nuclei. When thick hyphae were cultured on fresh medium, thin hyphae initially emerged, suggesting the necessity of sustained high metabolic activity.